# A Study on Site Selection for Regional Air Rescue Centers Based on Multi-Objective Jellyfish Search Algorithm

**DOI:** 10.3390/biomimetics8020254

**Published:** 2023-06-14

**Authors:** Yong Liao, Yiyang Zhao, Na Fang, Jie Huang

**Affiliations:** College of Air Traffic Management, Civil Aviation Flight University of China, Guanghan 618307, China

**Keywords:** aviation emergency rescue, multi-objective siting model, radiation function, multi-objective jellyfish search algorithm

## Abstract

In recent years, air emergency rescue capabilities have become increasingly important as an indicator of national comprehensive strength and development status. Air emergency rescue performs an indispensable role in addressing social emergencies by virtue of its fast response capabilities and extensive coverage. This vital aspect of emergency response ensures the timely deployment of rescue personnel and resources, enabling efficient operations in diverse and often challenging environments. To enhance regional emergency response capabilities, this paper presents a novel siting model that overcomes the limitation of single-objective approaches by integrating multiple objectives and considering the synergistic effects of network nodes, and the corresponding efficient solving algorithm is designed for this model. First, a multi-objective optimization function is established that fully incorporates the construction cost of the rescue station, response time, and radiation range. A radiation function is developed to evaluate the degree of radiation for each candidate airport. Second, the multi-objective jellyfish search algorithm (MOJS) is employed to search for Pareto optimal solutions of the model using MATLAB tools. Finally, the proposed algorithm is applied to analyze and verify the site selection for a regional air emergency rescue center in a certain region of China, and ArcGIS tools are used to draw the site selection results separately by prioritizing the construction cost under different numbers of site selection points. The results demonstrate that the proposed model can achieve the desired site selection goals, thus providing a feasible and accurate method for future air emergency rescue station selection problems.

## 1. Introduction

Aviation emergency rescue is the most timely and effective unconventional method for rescuing lives in response to emergencies and has become an important component of emergency rescue systems in developed countries and some developing countries. The level of aviation emergency rescue reflects a country’s emergency response rate and emergency aircraft assembly level when dealing with sudden events [1]. Due to its characteristics of short response time and few geographical limitations, aviation performs a very important role in emergency rescue activities. As an alternative takeoff and landing point for emergency rescue aircraft, aviation emergency rescue airports must be networked and grid-based, with hierarchical and type-based layouts [2]. Among all aviation emergency rescue stations, the regional aviation emergency rescue center is the highest-level rescue station, focusing on responding to high-level accidents with broad social impacts as its mission responsibilities. The rescue areas involve medical rescue, rescue material transportation, disaster prevention and reduction, and urban high-rise fire rescue within the administrative region. There are many factors that affect the regional aviation emergencies rescue center, such as existing supporting facilities around the airport, airport accessibility, geographical conditions, and airspace meteorological conditions.

Currently, the research on emergency rescue site selection can be divided into three categories according to the research objects [3]. The first category is the site selection of emergency medical stations, and the current research methods mainly include minimizing response time [4,5], maximizing treatment possibility [6], and maximizing coverage range [7]. The second category is the site selection of emergency logistics stations. The main research methods include a two-layer planning model considering material guarantee [8,9], spatial site selection decision-making based on geographic information system (GIS) analysis [10], and an improved site selection model combining dynamic and random selection [11]. The third category is the site selection of disaster prevention and relief stations. When dealing with emergency rescue, not only should the distribution of rescue materials and the setting of supporting equipment be considered, but also the needs of a large number of people in distress to be evacuated. The main methods include a mixed integer programming model that combines emergency rescue characteristics [12] and an evaluation model considering the convenience of public evacuation and transfer [13]. In addition to the above classical site selection models, there are also many models based on multi-objective optimization [14,15].

Aviation emergency rescue is characterized by a fast response, strong mobility, wide rescue range, good rescue effect, high technology content, strong rescue radiation, and short emergency response time. It is the most timely and effective means of organizing rescue and saving lives, with extremely broad application prospects. With flexible mobilization, many developed countries around the world regard improving aviation emergency rescue capabilities as the primary task of social security, such as the United States, which has established a complete aviation emergency rescue system [16], and Germany, which has 106 helicopters available for aviation emergency rescue. However, the development of China’s aviation emergency rescue system is still in the early stage. During the 14th Five-Year Plan period, the country issued a series of policies and documents to promote the construction of the aviation emergency rescue system. However, due to China’s vast territory, insufficient existing equipment, and uneven distribution of facilities and stations, it is urgently necessary to propose a reasonable theoretical method for selecting and locating aviation emergency rescue stations. At present, research on the selection and location of aviation emergency rescue stations is mostly qualitative analysis, such as the correlation analysis of various aviation emergency rescue standards [17], the evaluation of the level of an emergency rescue system for urban general aviation [18], the layout research of aviation emergency rescue sites in super-large cities using P-median model [19,20], and the research on the selection of the center location in civil aviation emergency rescue area using analytic hierarchy process (AHP) method [21], etc.

After conducting an analysis of current research on the selection of locations for aviation emergency rescue both domestically and internationally, it is evident that previous research on aviation emergency rescue facility location selection has predominantly focused on treating candidate nodes as independent entities, overlooking the synergistic effects within the rescue network. To address this research gap, this study introduces a vulnerability indicator to evaluate the network impact of nodes, considering their interconnectedness. Additionally, the prevalent adoption of single-objective location models by scholars may fail to capture the intricate complexities inherent in real-world scenarios. To overcome this limitation, we propose a multi-objective 0–1 programming model that incorporates the primary objective of emergency rescue station coverage, along with secondary objectives of construction costs and rescue response time. Moreover, this paper develops an efficient and rational solution algorithm tailored to the proposed location model, enhancing the decision-making process for aviation emergency rescue facility selection. To solve the Pareto optimal solution, the jellyfish search algorithm is applied. Furthermore, a case study is conducted to demonstrate the application of the proposed method and to enhance the fundamental theory behind aviation emergency rescue site selection.

## 2. Air Rescue Site

### 2.1. Definition of the Air Rescue Site

Aviation emergency rescue refers to a method of implementing rescue using aviation technology and equipment. Compared with other conventional emergency rescue methods, the unique feature of aviation emergency rescue is the use of rescue equipment with higher technological content. The subject implementing the rescue needs to undergo professional training and adhere to professional rescue principles.

Aviation emergency rescue stations are standby sites or landing points for implementing aviation emergency rescue tasks. They can be divided into regional aviation emergency rescue centers, urban aviation emergency rescue bases, and area aviation emergency rescue landing points according to their service function levels. Equipped aircraft can be fixed-wing aircraft or helicopters, including manned aircraft and unmanned aerial vehicles. The classification of aviation emergency rescue stations is shown in Figure 1.

### 2.2. Classification of the Air Rescue Site

According to the Beijing University of Aeronautics and Astronautics general aviation industry research center led the preparation of the “aviation emergency rescue station level classification standards” (group standards have been officially released) [2], we divided the aviation emergency rescue station into the regional air rescue center, city air rescue base and air rescue landing point three levels, where the relationship between the three is shown in Figure 2, and, respectively, specify the construction requirements of each level and facilities and equipment and staffing Standard. According to the set aviation emergency rescue objectives, the standard planning layout of aviation emergency rescue stations can be referred to.

#### 2.2.1. Regional Air Rescue Center

The regional air rescue center is a unit responsible for responding to a wide range of emergencies and providing medical relief at a high level across provinces. It is responsible for the rapid response, command and control, and implementation of emergency rescue missions in the region.

#### 2.2.2. City Air Rescue Base

The city air rescue base is a prefectural-level unit responsible for responding to and providing medical assistance for relatively low-level emergencies within its jurisdiction without affecting other cities. It is tasked with providing rapid response, command and control, and implementation of aviation emergency rescue missions on a city-wide scale.

#### 2.2.3. Air Rescue Landing Point

The distribution of air rescue landing points is primarily based on the radiation radius and response time of rescue helicopters, with the aim of establishing sites with suitable conditions for vertical takeoff and landing that can respond quickly to rescue tasks. The emphasis is on ensuring a uniform and widespread distribution of such sites throughout the region so that in the event of an unexpected rescue task, vertical takeoff and landing aircraft can be rapidly dispatched to the rescue site for the purpose of rescue.

### 2.3. Features of Regional Air Rescue Center

According to the “Classification Standards for Levels of Aviation Emergency Rescue Stations” [2], regional aviation emergency rescue centers have the following characteristics.

Regional aviation emergency rescue centers must be equipped with runways, aprons, hangars, command centers, office buildings, training centers, material warehouses, maintenance stations, small aviation parts warehouses, medical stations, and fire stations for use in executing aviation emergency rescue tasks. If these facilities already exist within an airport and their operation does not interfere with aviation emergency rescue functions, they may be used; otherwise, additional facilities must be constructed.Regional aviation emergency rescue centers require call centers, which may also utilize civil aviation airports adjacent to the rescue site or other relevant municipal call centers in the area.The types of aircraft commonly available at regional aviation emergency rescue centers are fixed-wing aircraft and helicopters. The number of rescue helicopters required in different regions will be determined based on demand, with a minimum of three for rescue purposes and two for training purposes, and at least one fixed-wing aircraft.The types of materials commonly available at regional aviation emergency rescue centers should meet the needs of implementing aviation emergency rescue tasks for public health emergencies, traffic emergencies, fire emergencies, earthquake emergencies, industrial emergencies, medical assistance, etc. The number of materials available at regional aviation emergency rescue centers will be determined based on the population of the region, with a minimum requirement of 0.0001% of the total population, multiplied by the corresponding multiple based on the type and frequency of use of the materials.Personnel will be allocated based on their job positions and the number of aircraft, divided into ground personnel and onboard personnel. Ground personnel will be allocated according to their job positions to meet minimum requirements, while on-board personnel will be allocated based on the operating requirements of the rescue aircraft and the number of aircraft used.

Considering the above characteristics of regional aviation emergency rescue centers, when selecting sites for these centers, priority should be given to civil aviation airports and A1 general airports in the central city of the region, where existing facilities and resources can be utilized, effectively reducing construction costs. Class A1 general aviation airports represent a subcategory within the broader classification of Class A general aviation airports. Specifically, Class A1 airports encompass those that facilitate commercial passenger flight activities and accommodate aircraft with a passenger seating capacity of 10 or more. On the other hand, Class A general aviation airports that are open to the public refer to those that grant public access for the purpose of obtaining flight services or engaging in individual flight operations.

## 3. Regional Air Rescue Center Site Selection Issues

### 3.1. Problem Description

According to the analysis above, aviation emergency rescue stations can be divided into three categories: regional aviation emergency rescue centers, urban aviation emergency rescue bases, and regional aviation emergency rescue take-off and landing points. Different types of aviation emergency rescue stations have different requirements and characteristics, and the factors influencing their location selection differ significantly, making it difficult to solve the location selection problem of the three types of aviation emergency rescue stations with a universal mathematical model. Therefore, it is reasonable to establish corresponding mathematical models for different types of aviation emergency rescue stations based on their characteristics. This paper mainly focuses on the location selection of regional aviation emergency rescue centers. Unless otherwise specified, the research object in the following text refers to regional aviation emergency rescue centers.

In essence, the location selection of regional aviation emergency rescue centers is the determination of the center’s location. Within a larger region (such as a country or a city cluster within a country), there is not only one suitable location for establishing an emergency rescue center. In order to save costs and maximize the effectiveness of emergency rescue, a minimum number of regional aviation emergency rescue centers should be established. The location of aviation emergency rescue centers has a decisive impact on the effectiveness of emergency rescue. The rational layout of aviation emergency rescue centers can save construction costs, enhance the radiation range of emergency rescue, and shorten response time. Therefore, the optimization of location selection for emergency rescue centers is of great significance.

### 3.2. Analysis of Influencing Factors

In the process of selecting the location for a regional aviation emergency rescue center, multiple perspectives need to be taken into consideration, such as construction cost and lead time, whether the site selection meets the long-term development needs of emergency rescue in the region, rescue response time for dealing with unexpected events, and the radiation range of the aviation emergency rescue center in the region. This article mainly focuses on the following three factors:

#### 3.2.1. Construction Cost Factor

Construction cost refers to the cost of establishing an aviation emergency rescue center, which is usually measured in currency and can also be expressed using other indicators. Considering the characteristics of emergency rescue operations, the selection of a node for a regional aviation emergency rescue center should be a permanent emergency facility that exists in a fixed location and can be quickly established or activated and provide emergency services in emergency situations. It belongs to the planning scope before large-scale emergency events occur. The site selection decision for a permanent emergency facility is based on the prediction and analysis of the situation of the events that may occur in order to provide timely basic emergency services and minimize losses. The construction cost factor mainly considers the supporting facilities around the airport, the geographical conditions of the airport, such as the existing medical, fire, and rescue institutions, and the level of resources available to support the normal operation of the airport.

#### 3.2.2. Radiation Factor

Radiation refers to the radiation range of the airport node in the region, which can be understood as the rescue ability of the aviation emergency rescue center that can be reached. Its influencing factors include the emergency accident propagation capability of the airport, the number of aircraft operations that the airport can handle within a certain period of time, and the linkage between the node and other nodes within the network.

#### 3.2.3. Response Time Factor

Response time refers to the time consumed by the aviation emergency rescue station from receiving the rescue information to arriving at the rescue destination and starting to perform the rescue task. Considering the special nature of emergency rescue operations, the shorter the rescue response time, the smaller the social losses caused by unexpected accidents. Therefore, the timeliness of the site selection results must be fully considered. Factors affecting timeliness include the accessibility of the airport’s land and air sides, the importance and impact of the airport area, and other factors.

## 4. Establishment of Site Selection Model for Regional Air Rescue Center

### 4.1. Mathematical Description of Site Selection Problem

Based on the analysis in Section 2.3, using existing civil transport airports as candidate sites for regional emergency rescue centers has the following advantages:It can make full use of the implementation equipment and various resources of existing civil transport airports, thus saving costs.China has a large number of civil transport airports, providing sufficient alternative options for the site selection of emergency rescue centers.China’s civil transport airports are located in densely populated, economically developed, or strategically important areas where there is a greater demand for emergency rescue, making it logical to establish emergency rescue centers in these areas.

In light of the above analysis, the mathematical description of the site selection for regional aviation emergency rescue centers is as follows:

In a certain region (which can be a small country in terms of land area or a metropolitan area in a larger country), there are *N* existing civil transport airports, and among these *N* airports, *n* (*n* < *N*) airports are selected as the regional aviation emergency rescue centers to be constructed in accordance with the requirements of regional aviation emergency rescue centers. The problem is: considering the factors mentioned in Section 3.2, which *n* airports should be selected as the most suitable emergency rescue centers?

### 4.2. Model Assumptions

**Assumption 1.** 
*The emergency response time is determined by the aviation accessibility time and does not consider the airport accessibility time. The accessibility time refers to the time consumed by various transportation modes to achieve a specified displacement.*


**Assumption 2.** 
*The construction cost of the airport rescue station only considers the construction cost based on the existing surrounding emergency rescue facilities and does not consider the cost of future expansion and construction of emergency rescue stations.*


**Assumption 3.** 
*The same airport can be covered by multiple regional aviation emergency rescue centers simultaneously, but after comparison, the center with the shortest response time among all the radiating aviation emergency rescue center points covering that airport is selected as the final center node for that site.*


**Assumption 4.** 
*The scope of application of each aviation emergency rescue center includes various emergency rescue functions, not targeting a specific type of rescue, and involves medical rescue, disaster reduction, relief, urban high-altitude firefighting rescue, etc., within the region.*


### 4.3. Definition of Decision Variables

The present model defines two binary decision variables, *x_i_*, and *y_ij_*, with values of 0 or 1. The variable *y_ij_* represents whether or not node *i* can radiate node *j*, while *x_i_* indicates whether or not node *i* is selected as the radiating center node.
(1)$$y_{ij}=\left\{\begin{array}{l} 1,\;\mathrm{if}\;\mathrm{the}\;\mathrm{radiation}\;\mathrm{point}\;j\;\mathrm{can}\;\mathrm{be}\;\mathrm{reached}\;\mathrm{by}\;\mathrm{the}\;\mathrm{center}\;\mathrm{point}\;i\\ 0,\;\mathrm{if}\;\mathrm{the}\;\mathrm{radiation}\;\mathrm{point}\;j\;\mathrm{can}\;\mathrm{not}\;\mathrm{be}\;\mathrm{reached}\;\mathrm{by}\;\mathrm{the}\;\mathrm{center}\;\mathrm{point}\;i \end{array}\right.\;$$
(2)$$x_{i}=\left\{\begin{array}{l} 1,\;\mathrm{if}\;\mathrm{the}\;\mathrm{central}\;\mathrm{airport}\;\mathrm{point}\;i\;\mathrm{is}\;\mathrm{selected}\;\\ 0,\;\mathrm{if}\;\mathrm{the}\;\mathrm{central}\;\mathrm{airport}\;\mathrm{point}\;i\;\mathrm{is}\;\mathrm{not}\;\mathrm{selected}\; \end{array}\right.$$

### 4.4. Establishment of Objective Function

#### 4.4.1. Construction Cost Objective

To minimize the construction cost of aviation emergency rescue stations, the present model proposes a minimum construction cost objective function, where *A_i_* represents the construction cost coefficient of node *i*. A higher coefficient indicates a higher construction cost.
(3)$$\min Z_{1}=\sum_{i=1}^{n}A_{i}x_{i}$$

#### 4.4.2. Response Time Objective

The response time coefficient *B_i_* refers to the time it takes for an aviation emergency rescue station to receive rescue information, dispatch rescue aircraft, and arrive at the rescue destination. It is calculated as the ratio of the maximum response time within the radiation range of node *i* to the number of rescue stations in the selected scheme. Considering the unique nature of aviation emergency rescue missions, the present model proposes the shortest response time objective function to improve the efficiency of rescue operations and minimize the losses and damages caused by emergencies.
(4)$$\min Z_{2}=\sum_{i=1}^{n}B_{i}x_{i}$$
(5)$$B_{i}=T_{i}+T_{R}$$
(6)$$T_{i}=\max\left\{t_{ij}y_{ij}\right\}\;i,j\in V\;\mathrm{and}\;i\neq j$$
where *T_i_* represents the maximum response time that can be radiated within the radiation range of node *i*, *T_R_* represents the flexible time for emergency rescue maneuvers, including rescue organization and force gathering time, *t_ij_* represents the rescue response time from node *i* to node *j*, *V* represents the set of nodes to be selected.

However, if only these two parameters are considered, the impact of the number of selected sites on the response time cannot be reflected. Therefore, based on Equation (7), this paper proposes the use of the following formula to calculate the response time coefficient. As the number of selected sites increases, the response time coefficient decreases, which is more in line with real rescue situations.
(7)$$B_{i}=\frac{T_{i}+T_{R}}{\sum_{i=1}^{n}x_{i}}\;i\in V$$

#### 4.4.3. Radiation Degree Objective

Radiation degree coefficient

The radiation radius refers to the coverage radius of the rescue aircraft that performs aviation emergency rescue tasks at the aviation emergency rescue station, which is the farthest rescue point that the rescue aircraft can reach within a specified time from the station center. In the basic site selection model used to solve location problems, the maximum coverage location model is a commonly used single-objective model based on on-site coverage, where the coverage is defined as a binary 0–1 variable, with node *i* being covered as 1 and not fully covered as 0. Following the approach of the maximum coverage model to address the site selection problem of the regional aviation emergency rescue center, if the actual distance from the rescue station to the accident site is greater than the radiation radius or the response time exceeds the standard limit, the emergency rescue service cannot be included in the radiation range of the candidate points, which does not meet the actual rescue needs. Therefore, the radiation degree evaluation in this paper is based on the maximum coverage model, introducing a radiation range evaluation index, i.e., extending the radiation degree as a function between [0, 1], whose value is determined by the response efficiency from airport *i* to the candidate regional aviation emergency rescue center. This paper characterizes the radiation degree coefficient *τ_ij_* as the reachable time between airport *i* and the candidate rescue station and represents it using a piecewise inverse cosine function, thereby achieving the goal of extending the multivariate radiation degree.
(8)$$\tau_{ij}=\left\{\begin{array}{l} 0,\;t_{ij}\geq t_{\max}\\ \frac{2}{\pi }\arccos\left(\frac{t_{ij}-t_{\min}}{t_{\max}-t_{\min}}\right),\\ 1,\;t_{ij}\leq t_{\min} \end{array}\right.t_{\min}\leq t_{ij}\leq t_{\max}\;i,j\in V\;\mathrm{and}\;i\neq j$$
where *τ_ij_* represents the radiation degree coefficient from the central node *i* to the radiation node *j*, *t*_max_ represents the maximum responsive time, i.e., the radiation radius set by the model, and *t*_min_ represents the critical value of effective response time. If the response time is less than this value, it is considered that the rescue aircraft can easily cover it, and its radiation degree coefficient value is 1.

2.Vulnerability coefficient

This model defines the vulnerability coefficient *ε_i_* as the degree of the decrease in the emergency rescue network’s network rescue capability when node *i* in the emergency rescue network experiences a sudden accident and cannot operate normally, reflecting the degree of loss and damage that may occur when node *i* experiences a sudden accident. When the vulnerability coefficient is larger, it indicates that the emergency rescue capability is stronger and the risk of suffering disaster losses is smaller when a sudden event occurs, thus having a smaller negative impact on the regional network. Considering the universality of aviation emergency rescue tasks and factors, such as population distribution, economic conditions, transportation, and multiple coverages of some key areas, this model constructs the maximum radiation function:(9)$$\max Z_{3}=\sum_{i=1}^{n}\varepsilon_{i}\sum_{j=1}^{n}\tau_{ij}y_{ij}$$

Using the radiation results of the regional air emergency rescue center selection in the area shown in Figure 3 as an example, the yellow nodes represent radiation points, and the red nodes represent center points. Assuming that the fragility coefficients of the three center nodes are 0.836, 0.545, and 0.326, the lines between the center points and the radiation points represent the degree of radiation coverage (specific data can be found in Table 1), and their thickness represents the response level, i.e., the product of the fragility coefficient of the center point and the radiation coefficient. The radiation function result for this location selection is calculated as follows: the objective response time for this selection result is 37.11.

### 4.5. Constraint Construction

To ensure the full coverage of all nodes, i.e., radiation node *j* can be covered by one or more center nodes, let *V* denote the set of candidate airport nodes. Therefore, the following constraint formula is proposed in this model:(10)$$\sum_{i=1}^{n}y_{ij}\geq 1\;i,j\in V\;\mathrm{and}\;i\neq j$$

To ensure the effectiveness of the regional aviation emergency rescue center, i.e., center node *i* has at least one node *j* that can be radiated, where *i ≠ j*, the following constraint formula is proposed in this model. The schematic diagram of this constraint is shown in Figure 4.
(11)$$x_{i}\left(\sum_{j=1}^{n}y_{ij}-1\right)\geq 0\;i,j\in V\;\mathrm{and}\;i\neq j$$

The values of the two decision variables must satisfy the following formula relationship, that is, only when node *i* is selected as the radiation center, whether center node *i* can radiate node *j* will be considered. On the contrary, if node *i* is not selected, the value of *y_ij_* will not be considered. The relationship between the values of the two decision variables is shown in Figure 5:(12)$$y_{ij}\leq x_{i}\;i,j\in V\;\mathrm{and}\;i\neq j$$

To ensure the effectiveness of the model, i.e., at least one Regional Aviation Emergency Rescue Center point exists, the following constraints are proposed:(13)$$\sum_{i=1}^{n}x_{i}\geq 1\;i\in V$$

To ensure the rationality of the economic and time investment in the early stage of construction, the following constraints are proposed:(14)$$\sum_{i=1}^{n}x_{i}\leq M\;i\in V$$
where *M* represents the maximum number of regional air rescue centers that can be selected, which is determined by relevant departments considering various factors, such as airport operation conditions, socio-economic population conditions, natural disaster occurrence rates, and cooperation among airports in the region.

To ensure the timeliness of rescue response time and consider more radiation nodes, the maximum radiated response time is less than the radiation radius time, and the following constraint function is established:(15)$$t_{ij}y_{ij}<t_{\max}\;i,j\in V\;\mathrm{and}\;i\neq j$$

In summary, the general flowchart of the method of changing the thesis is shown in the Figure 6. The model for the selection of the Regional Aviation Emergency Rescue Center proposed in this paper is:
Figure 6Regional aviation emergency rescue center site selection method flowchart.
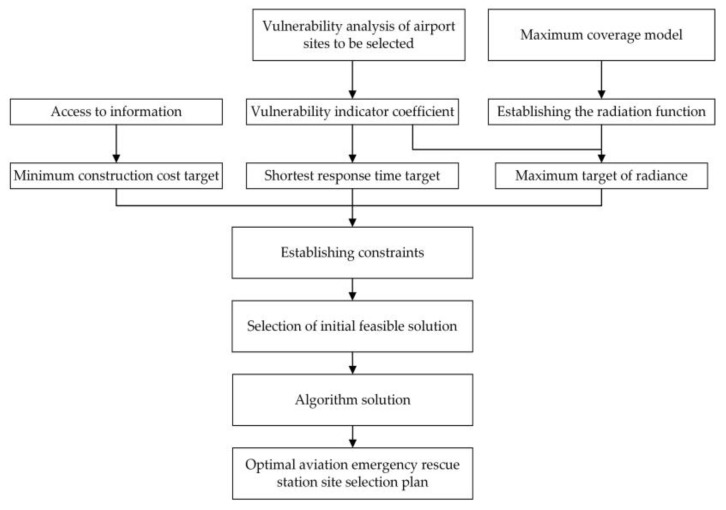




(16)
$$\begin{array}{ll} obj. & \min Z_{1}=\sum_{i=1}^{n}A_{i}x_{i}\\ & \min Z_{2}=\sum_{i=1}^{n}B_{i}x_{i}\\ & \max Z_{3}=\sum_{i=1}^{n}\varepsilon_{i}\sum_{j=1}^{n}\tau_{ij}y_{ij} \end{array}$$


(17)
$$s.t\left\{\begin{array}{l} x_{i}\geq y_{ij}\;i,j\in V\;\mathrm{and}\;i\neq j\\ \sum_{i=1}^{n}y_{ij}\geq 1\;i,j\in V\;\mathrm{and}\;i\neq j\\ x_{i}\left(\sum_{j=1}^{n}y_{ij}-1\right)\geq 0\;i,j\in V\;\mathrm{and}\;i\neq j\\ \sum_{i=1}^{n}x_{i}\leq M\;i\in V\\ \sum_{i=1}^{n}x_{i}\geq 1\;i\in V\\ t_{ij}y_{ij}<t_{\max}\;i,j\in V\;\mathrm{and}\;i\neq j\\ B_{i}=\frac{T_{i}+T_{R}}{\sum_{i=1}^{n}x_{i}}\;i\in V\\ T_{i}=\max\left\{t_{ij}y_{ij}\right\}\;i,j\in V\;\mathrm{and}\;i\neq j\\ x_{i},y_{ij}=0\;\mathrm{or}\;1\;i,j\in V\;\mathrm{and}\;i\neq j \end{array}\right.$$



## 5. Model Solution

### 5.1. Methods for Solving Multi-Objective Problems

When solving a multi-objective optimization model, there are usually two types of methods to choose from, namely, transformed single-objective optimization methods and intelligent heuristic algorithms. The model proposed in this paper exhibits the following characteristics:The model is nonlinear, which can be observed from Equations (7) and (11).The decision variables *x_i_* and *y_ij_* are high-dimensional variables.The time complexity of an algorithm is a measure of the amount of time required for the algorithm to solve a problem of a given size. In the case of an n-dimensional 0–1 vector, the time complexity of the algorithm is *O*(2*^n^*), which is a characteristic of exponential time complexity. In contrast, polynomial time complexity implies that the running time of the algorithm grows polynomially with the size of the input, making it more efficient than exponential time complexity. Despite the algorithm’s exponential time complexity, it is still possible to verify a feasible solution within polynomial time. Therefore, the problem addressed in this paper is classified as an NP-hard problem. NP-hard problems are a class of problems that are known to be difficult to solve and are thought to require exponential time complexity.

Given these characteristics, transforming the multi-objective problem into a single-objective problem to find the absolute optimal solution quickly [14] is a suitable method for ideal cases, but it may not be adequate for the problem addressed in this paper. Therefore, the model proposed in this paper cannot be solved by converting the problem into a single objective. With the development of computers, various intelligent optimization algorithms have been proposed. Intelligent heuristic algorithms use self-defined evaluation methods to iteratively search the state space until finding the Pareto optimal solution. Currently, common algorithms include genetic algorithms [22,23], taboo search algorithms [24,25], and particle swarm algorithms [26,27,28]. Genetic algorithms, as a widely used heuristic algorithms, have many variants. Among them, the non-dominated sorting genetic algorithm II (NSGA-II) is currently the mainstream algorithm for solving multi-objective problems [29].

The artificial jellyfish search (JS) optimizer algorithm, proposed by Jui-Sheng Chou et al. from Taiwan in 2020, mainly simulates the characteristics of jellyfish floating with the ocean current and internal movement within the jellyfish population and introduces a time control mechanism [30]. The advantages of this algorithm are as follows:It has only two internal parameters;It is easy to encode;It is easy to apply [15,31];JS can search for the optimal position better than other algorithms;It requires less time and has a faster convergence rate than other algorithms [32];In large-scale mathematical functions, the significance of the Wilcoxon rank sum test further confirms JS’s ability to find the optimal value in large-scale functions [33,34,35];JS is significantly better than the firefly Algorithm (FA), gravitational search algorithm (GSA), artificial bee colony algorithm (ABC), differential evolution (DE), particle swarm optimization (PSO), and genetic algorithm (GA) in mathematical benchmark tests [30].

The JS algorithm has achieved good results in comparative experiments and therefore has a wide range of applications. The jellyfish search algorithm is a particle-like biomimetic intelligent algorithm that optimizes based on jellyfish population migration and predation behavior. Since the distribution of jellyfish food in the ocean is uneven, jellyfish will move in the direction of the area with relatively more food during migration. This algorithm mathematically models the above behavior of jellyfish. In actual optimization models, each jellyfish represents a solution, and the function that measures the relative quantity of food in the sea is called the fitness function. The flowchart of the jellyfish search algorithm for model solving is shown below (Figure 7), and the steps for model solving are as follows:Initialization of population generation;Determination of the initial optimal position;Updating time control parameter *C(t)*;Updating jellyfish positions based on the direction of ocean currents;Updating the movement types of individuals based on their movement type;Evaluating new fitness and updating the optimal position;Evaluating the fitness of the latest jellyfish position;Determining whether the maximum number of iterations has been reached.

**Figure 7 biomimetics-08-00254-f007:**
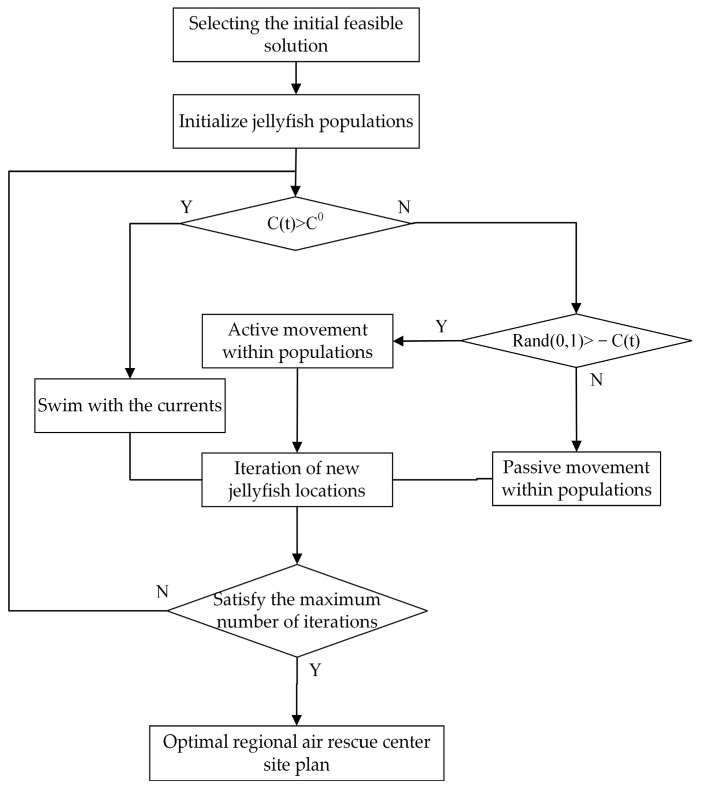
Jellyfish search algorithm flowchart.

### 5.2. Algorithm Design

#### 5.2.1. Population Initialization

The initial population of jellyfish is formed. Assuming that the jellyfish is normally distributed in all dimensions, to avoid the problem of premature convergence, the algorithm uses the logistic chaos mapping method to replace the normal random initialization. The logistic chaos mapping method is used to generate *N_pop_* data strings *J*, where the *i*-th data string is used to represent the *i*-th individual jellyfish, and they form a population.
(18)$$J_{i}={\left(\begin{array}{cccc} x_{1} & x_{2} & \cdots & x_{i} \end{array}\right)}_{1\times v}^{T}$$
(19)$$J_{0}=J_{\min}+rand\left(J_{\max}-J_{\min}\right)$$
(20)$$J_{i+1}=\eta J_{i}\left(1-J_{i}\right),\;0\leq J_{0}\leq 1$$
where *J_i_* is the logistic chaos value of the *i*-th jellyfish position; *J*_0_ is used to determine the initial jellyfish population position and referring to the original algorithm parameter setting [33,34], $J_{0}\notin \left\{0,\;0.25,\;0.75,\;1\right\}$; parameter *η* = 4; *J*_min_ and *J*_max_ represent the minimum and maximum values of the search space, respectively.

The individuals generated by the mapping are iterated from the initial values, and in this paper, the population size is set to 200. Each individual in the initial population is subjected to a constraint check one by one. Random individuals that do not satisfy the constraints in the model are eliminated through a fitness penalty item. The more constraint conditions are not satisfied, the greater the probability of elimination. The eliminated individuals are replaced with new initial individuals until the population size is met.

#### 5.2.2. Time Control Mechanism

There are mainly two forms of jellyfish foraging behavior in the simulated ocean, i.e., following the ocean current and active or passive movement within the jellyfish population. A time function is introduced to control different types of jellyfish movement and analyze their time characteristics according to the number of search iterations.
(21)$$C\left(t\right)=\left|\left(1-\frac{t}{t_{\max}}\right)\left(2\times rand\left(0,1\right)-1\right)\right|$$
where *C(t)* represents the time control function. If *C(t)* > 0.5, the jellyfish follows the ocean current, and if *C(t)* < 0.5, the jellyfish moves within the population. *t* represents the search iteration.

#### 5.2.3. Boundary Constraint Mechanism

For the phenomenon of crossing boundaries, considering that the ocean is distributed around the world and the jellyfish search algorithm is based on the spherical structure of the earth, a boundary buffering strategy is adopted. That is when the jellyfish exceeds the search range, it will return to the opposite boundary. The iteration diagram is shown in Figure 8.
(22)$$\left\{\begin{array}{l} J_{i,d}^{\prime }=\left(J_{i,d}-U_{b,d}\right)+J_{\min}\left(d\right),\;if\;J_{i,d}>J_{\max,d}\\ J_{i,d}^{\prime }=\left(J_{i,d}-L_{b,d}\right)+J_{\max}\left(d\right),\;if\;J_{i,d}<J_{\min,d} \end{array}\right.$$

#### 5.2.4. Optimization Search Stage

The jellyfish search algorithm’s optimization search mechanism divides the search into three types: flow-following search, passive individual search within the population, and active individual movement search.

Flow-following search

Considering the impact of food-rich ocean currents on jellyfish, the direction of the ocean current is used to represent the mean difference between the optimal jellyfish position and the population jellyfish position. Assuming that the jellyfish position follows a normal distribution, the new jellyfish position can be calculated by the following formula:(23)$$J_{i}^{\left(t+1\right)}=J_{i}^{\left(t\right)}+rand\left(0,1\right)\times \vec{D}$$
(24)$$\vec{D}=J^{*}-\beta \times rand\left(0,1\right)\times \frac{e_{C}\sum J_{i}}{N_{pop}}$$
where β = 3 is the distribution coefficient, *J^*^* represents the current optimal jellyfish position, *N_pop_* represents the population size, and *e_C_* is the attraction control factor.

A large number of nutrients in the ocean current will attract jellyfish, thereby changing their direction of movement. The algorithm’s convergence trend is bio-mimetically derived from the direction of the ocean current, which is determined by calculating the average vector position of each jellyfish to the current optimal jellyfish position:(25)$$\vec{Tr}=\frac{1}{N_{pop}}\sum {\vec{Tr}}_{i}$$
(26)$${\vec{Tr}}_{i}=\left(J^{*}-e_{c}J_{i}\right)$$
(27)$$\mathrm{Set}\begin{array}{cc} & df{=e}_{c}\mu \end{array}df$$
where *N_pop_* is the size of the jellyfish population, *J^*^* is the position of the optimal jellyfish, *e_C_* is the attraction control factor, *μ* is the average position of all jellyfish, which is the difference between the optimal jellyfish position and the average position of all jellyfish. The new formula for the direction of the ocean current can be derived from the above equation:(28)$$\vec{Tr}=J^{*}{-e}_{c}\frac{\sum J_{i}}{N_{pop}}=J^{*}{-e}_{c}\mu$$

2.Population movement

Passive individual movement within the population. This method is the primary movement method of jellyfish in the early stage. If *C(t)* < 0.5 and *rand*(0, 1) ≥ 1 − *C(t)*, the formula for passive movement of jellyfish is:(29)$$J_{i}^{\left(t+1\right)}=J_{i}^{\left(t\right)}+\iota \times rand\left(0,1\right)\left(J_{\max}-J_{\min}\right)$$
where *J*_max_ and *J*_min_ represent the upper and lower limits of the search space, respectively, and *ι* = 0.1 is the motion coefficient.

Active individual movement within the population. This method is the primary movement method of jellyfish in the later stage. Jellyfish *i* can swim towards jellyfish *j* with more food, and if the amount of food at *j* is less than that of *i*, jellyfish *i* will swim away from *j*. If *C(t)* < 0.5 and *rand*(0, 1) < 1 − *C(t)*, the formula for passive movement of jellyfish is:(30)$$J_{i}^{\left(t+1\right)}=J_{i}^{\left(t\right)}+rand\left(0,1\right)\times \vec{Op}$$
(31)$$\vec{Op}=\left\{\begin{array}{l} J_{j}^{\left(t\right)}-J_{i}^{\left(t\right)},\begin{array}{cc} if & f\left(J_{i}^{\left(t\right)}\right)>f\left(J_{j}^{\left(t\right)}\right) \end{array}\\ J_{i}^{\left(t\right)}-J_{j}^{\left(t\right)},\begin{array}{cc} if & f\left(J_{j}^{\left(t\right)}\right)>f\left(J_{i}^{\left(t\right)}\right) \end{array} \end{array}\right.$$

#### 5.2.5. Update Location Stage

The fitness value of the initial individuals is evaluated based on an evaluation function *fit* to determine the quality of the individuals. The evaluation function used in this paper is as follows. The individual with the best fitness is used as the initial optimal position the paper takes $\lambda =\left(\begin{array}{ccc} 1 & 1 & -1 \end{array}\right)$, for the convenience of calculation, and introduces a penalty term *Θ*, so that the individuals that do not satisfy the constraints do not enter the fitness evaluation process.
(32)min fit=λ·Z+θ
(33)$$Z={\left(\begin{array}{c} A•J\\ B•J\\ Z_{3} \end{array}\right)}_{3\times 1}$$
(34)θ=pL
where $A={\left(\begin{array}{ccc} A_{1} & \cdots & A_{i} \end{array}\right)}_{1\times v}$ and $B={\left(\begin{array}{ccc} B_{1} & \cdots & B_{i} \end{array}\right)}_{1\times v}$ are matrices, and *p* is a penalty coefficient. If a jellyfish individual does not satisfy any of the constraint conditions, a penalty of one unit is added to the coefficient. Therefore, the penalty coefficient *p* has a range [0, 5], and L is the maximum penalty base, which is set to 10^9^ in this paper.

After evaluating the fitness of all individuals in the population, the individual with the best fitness is selected as the initial optimal location. The location of this individual is stored in the optimal location set. After updating the optimal location of the individual, the maximum number of iterations is checked. If the maximum number of iterations has not been reached, the time control parameters are updated, and the optimal location is searched again for the next iteration. If the maximum number of iterations has been reached, the optimal location and the optimal selection plan are output.

## 6. Case Study

### 6.1. Case Background

Based on the existing layout of civil aviation airports, it can be concluded that there are sufficient candidate transportation airport points for regional aviation emergency rescue centers in a certain region. Therefore, this paper selects the transportation airport in this region for validation of the regional aviation emergency rescue center location. The same calculation process and different parameter settings can also be used to select suitable city aviation emergency rescue bases and area aviation emergency rescue landing points within the region. In this paper, this region is taken as an example. The region consists of three parts: A, B, and C, and there are 42 existing civil transportation airports, as shown in Figure 9. The area where V_1_–V_15_ is located is A, the area where V_16_–V_26_ is B, and the area where V_27_–V_42_ is C in the figure. The size of the airport node indicates the size of the airport’s flight area level.

### 6.2. Model Parameter Determination

#### 6.2.1. Construction Cost Coefficient

This paper employs a method of estimating construction cost coefficients for candidate sites based on the airport flight area class. After consulting relevant departments and experts from the industry, the maximum number of potential regional aviation emergency rescue centers is determined to be 12, taking into account construction costs. The specific estimation results of the maximum construction cost coefficient *A_i_* are presented in Table 2.

#### 6.2.2. Fragility Coefficient

The fragility coefficient of the air emergency rescue station node refers to the degree of loss and damage that may occur when the airport node experiences a sudden accident, reflecting the degree of the decline in the rescue network’s capabilities when one or more nodes in the emergency rescue network cannot operate normally due to a sudden accident. This paper’s fragility coefficient is obtained through the evaluation by the relevant expert group, and the specific values of *ε_i_* are shown in Table 2.

#### 6.2.3. Radiation Degree

To simplify the response time calculation, this paper uses the Euclidean distance of the airport node coordinates after converting the longitude and latitude of each airport node using ArcGIS. The average speed of various types of aircraft and the rescue accessibility speed is set to *v* = 500 Km/h. The optimal response time for air emergency rescue is within 1 h. Therefore, this paper proposes the maximum radiation range threshold value *t*_max_ = 35 min, and the maneuverable and flexible time is set to 30 min. When the response time is less than 10 min, the node is considered to be completely covered, i.e., *t*_min_ = 10 min. The response time is then substituted into Equation (7) for calculation. Taking area A as an example, the specific results are shown in Table 3, where V*i* represents the *i*-th candidate airport.
(35)$$\rho_{ij}=\sqrt{{\left(x_{j}-x_{i}\right)}^{2}+{\left(y_{j}-y_{i}\right)}^{2}}$$
where the longitude and latitude coordinates of the central node *i* are (*x_i_*, *y_i_*), and the longitude and latitude coordinates of the radiation node *j* are (*x_j_*, *y_j_*).

### 6.3. Multi-Objective Jellyfish Search Algorithm for Site Selection

#### 6.3.1. Algorithm Parameter Setting

Utilizing the mathematical model and algorithm design developed in this study, the problem solution is coded using MATLAB 2021a. Choosing appropriate parameters for the algorithm is critical for balancing search efficiency and search scope. If the parameters are set too large, it can slow down the iterative search process and result in an excessively large search space. Conversely, if the parameters are set too small, it can lead to insufficient search depth and reduced iterative efficiency. Based on expert opinions and the economic and demographic conditions of the region, this study selected a population size of *N_p__op_* = 200 and a maximum iteration count of 200. The iterative search process was terminated upon reaching the 200th iteration.

#### 6.3.2. Algorithmic Solution Search and Update

With a search for optimal siting results of regional aviation emergency rescue centers under various siting points within a specific region, the Pareto set was obtained after conducting 200 iterations using MATLAB software. Following the screening process, duplicate and non-prevailing siting solutions were eliminated. The resulting Pareto solution set was then plotted, and the specifics are illustrated in Figure 10. The x, y, and z axes in the graph represent the construction cost, response time, and radiance of each siting solution, respectively. Furthermore, the size of the red dots on the figure indicates the number of siting points, ranging from 6 to 11. The corresponding data of the Pareto solution set for different siting points are presented in Table 4, where Z1, Z2, and Z3 represent the construction cost, response time, and radiance of each siting solution, respectively.

#### 6.3.3. Example Analysis of Site Selection Results

The results obtained from solving Table 4 are presented in Figure 11a, which displays the variation of the objective function values for each siting solution. The different colors represent different numbers of siting points, while the different point styles indicate different objective function values. The *x*-axis represents the different siting solutions, while the *y*-axis represents the construction cost, response time, and radiance of each siting solution, respectively. The symbol size in the figure represents the number of siting points, ranging from 5 to 11. Figure 11b is generated based on the results obtained from Table 4, which displays the number of Pareto solutions for different numbers of siting points. The *x*-axis represents the different numbers of siting points, while the *y*-axis represents the number of solutions for each siting point. The labels in the figure represent the specific number of solutions for each number of siting points.

To visually demonstrate the radiation effect of the site selection scheme, this study presents the results of site selection for a regional aviation emergency rescue center with different numbers of site selection points, prioritizing construction cost. As only one Pareto solution set exists for five site selection points, an ArcGIS tool is employed to generate a radiation effect map on the map of the region. The radiation effect of site selection is illustrated in Figure 12, where the regional aviation emergency center point is indicated by the five-pointed star, and the yellow dots represent the radiation points. Specific siting results data are presented in Table 5 which includes the radiation intensity *l_i_*, i.e., the product of vulnerability and radiation degree, to simplify the radiation results. Optimal site selection for the five sites is determined by selecting the center that radiates a node the most efficiently. Based on the results, the optimal site selection for the five sites under the priority of construction cost is airport 1, 10, 21, 29, 34 as the regional aviation emergency rescue center, with corresponding target values of 8.5 for construction cost, 3.109603 for response time, and 5.73 for radiance.
(36)$$l_{i}=\varepsilon_{i}\times \tau_{ij}$$

To facilitate visualization, MATLAB software is utilized to plot the remaining site selection points from 6 to 11, respectively, with priority given to the construction cost of the site selection plan based on latitude and longitude coordinates, as illustrated in Figure 12. The pentagram represents the regional aviation emergency center point, while the line thickness li denotes the radiation intensity, i.e., the vulnerability and radiance product. For nodes that are radiated by multiple centers, the optimal center is selected, and different colors indicate the radiation range of different centers. The vertical and horizontal coordinates indicate the latitude and longitude coordinate values, respectively. When there are multiple siting schemes with the least construction cost, the maximum radiance target is considered, with the order of priority of the three targets being Z1, Z3, and Z2. Based on the results in Table 4, when selecting six centers, the Pareto solution set comprises a total of four site selection schemes, with Scheme 1 having construction cost, response time, and radiance target values of X, Y, and Z, respectively. The corresponding radiation latitude and longitude coordinates are presented in Figure 13a. The remaining Figure 13b–f, display the latitude and longitude coordinates of the site selection options for 7, 8, 9, 10, and 11 site selection points, respectively, obtained using MATLAB software.

## 7. Conclusions

The siting of aviation emergency rescue centers is crucial for enhancing emergency rescue capabilities, reducing construction costs, and minimizing social losses. To address this issue, this paper proposes a siting model for a regional aviation emergency rescue center based on a multi-objective 0–1 optimization model. The key contributions of this paper are as follows.

The paper proposes a solution approach that considers the construction cost, response time, and radiance objectives among the programs that meet the constraints of full coverage and siting points.The paper establishes a mathematical siting model using a multi-objective 0–1 optimization model, vulnerability coefficient, construction cost coefficient, and radiance coefficient. The model aims to minimize the construction cost and response time, maximize regional radiance as objective functions, and take different siting options as decision variables, and full coverage, central point effectiveness, and the number of siting points as constraints.The paper designs a multi-objective jellyfish search algorithm to solve the developed model, which is a multi-objective nonlinear optimization model and an NP-hard problem. The algorithm is designed to solve the model by initializing the population, time control mechanism, boundary restriction mechanism, merit-seeking search, and update position.The paper selects a region for the case study of the siting model. Through the multi-objective jellyfish search algorithm, the corresponding Pareto solution sets for different site selection points are solved, and the results demonstrate that the proposed method is well-applied to the regional aviation emergency rescue center siting problem.

However, the paper has limitations that can be improved in future research. For instance, the construction cost and access time coefficients used in the model have some defects due to inadequate data sources. While the current study assumes equal ability and experience for all rescuers due to space limitations, it is acknowledged that considering the quality, experience, and recent activity of rescuers in the crew rotation is a critical factor in optimizing rescue operations. Future research can explore the integration of these factors to enhance the practical applicability and realism of the proposed approach. Moreover, the heuristic algorithm used to solve the multi-objective nonlinear optimization problem may not always produce optimal solutions, and thus, future research should focus on verifying the global search ability of the algorithm. Overall, the proposed siting model and jellyfish search algorithm are valuable contributions to the field of aviation emergency rescue center siting and can be extended to other levels of aviation emergency rescue siting problems.

## Figures and Tables

**Figure 1 biomimetics-08-00254-f001:**
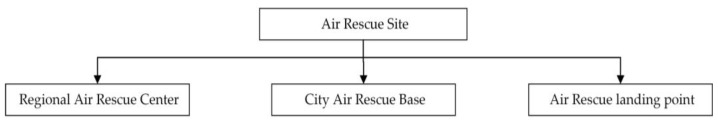
The basic structure of the aviation emergency rescue station.

**Figure 2 biomimetics-08-00254-f002:**
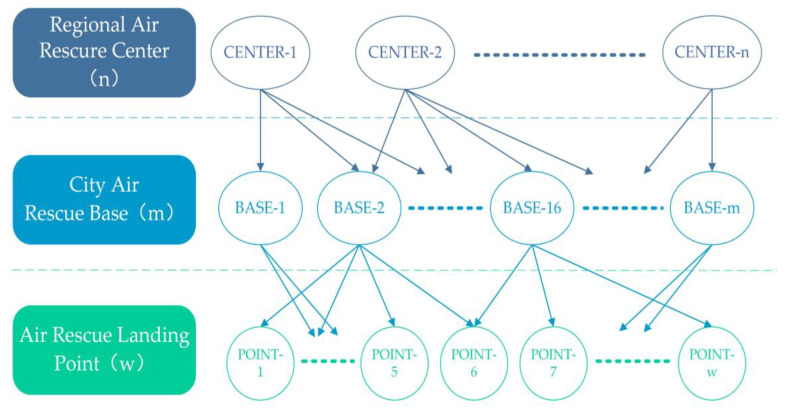
Schematic diagram of the network of aviation emergency rescue sites.

**Figure 3 biomimetics-08-00254-f003:**
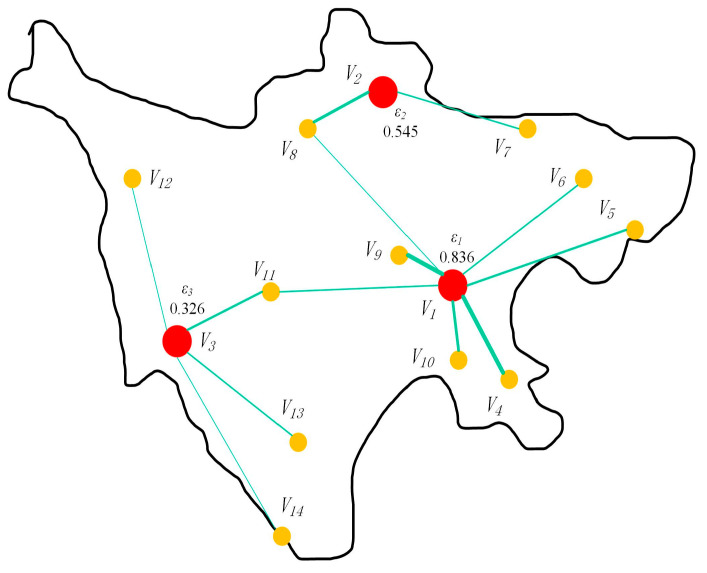
Schematic diagram of the location of the regional air emergency rescue center in a certain area.

**Figure 4 biomimetics-08-00254-f004:**
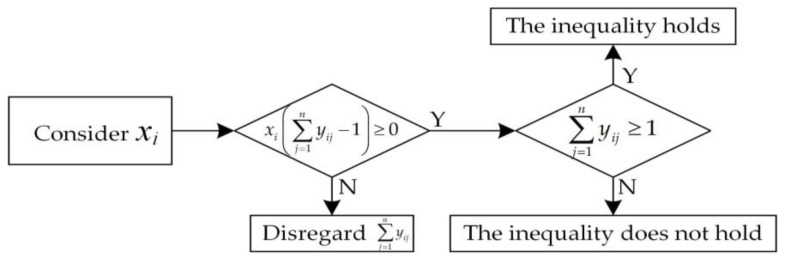
Schematic diagram of the effectiveness constraint of center nodes.

**Figure 5 biomimetics-08-00254-f005:**
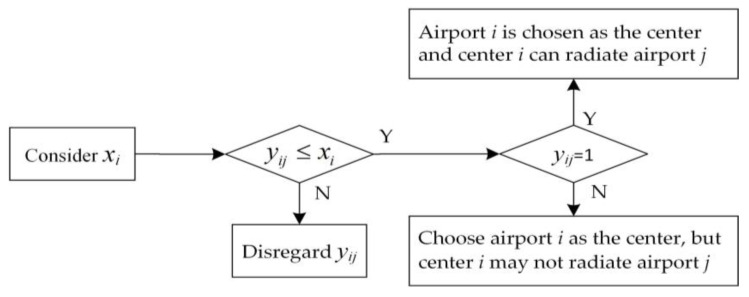
Schematic diagram of the relationship between the values of decision variables.

**Figure 8 biomimetics-08-00254-f008:**
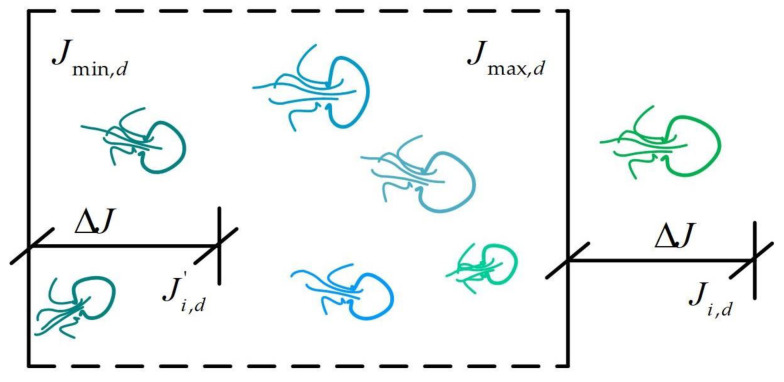
Flowchart of the jellyfish search algorithm.

**Figure 9 biomimetics-08-00254-f009:**
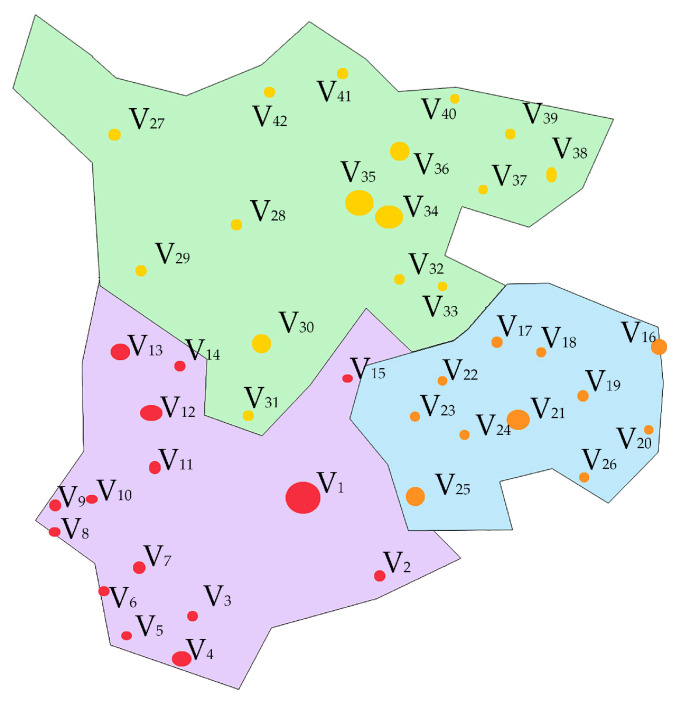
Distribution of civil airports in a region. Where different colors indicate different areas.

**Figure 10 biomimetics-08-00254-f010:**
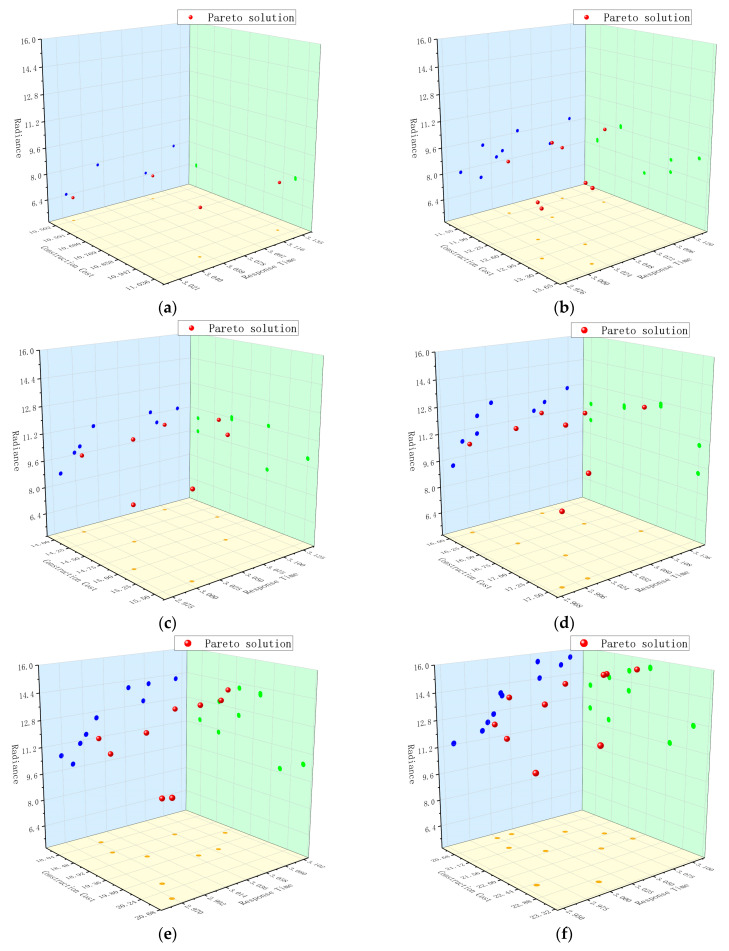
A total of 200 iterations of different siting points Pareto solution diagram. Where (**a**–**f**) denote, respectively, the Pareto solution set of the siting scheme for the cases of 6, 7, 8, 9, 10, and 11 siting points obtained by processing with MATLAB software. where the points of different colors indicate the distribution on the projection plane of the corresponding objective function.

**Figure 11 biomimetics-08-00254-f011:**
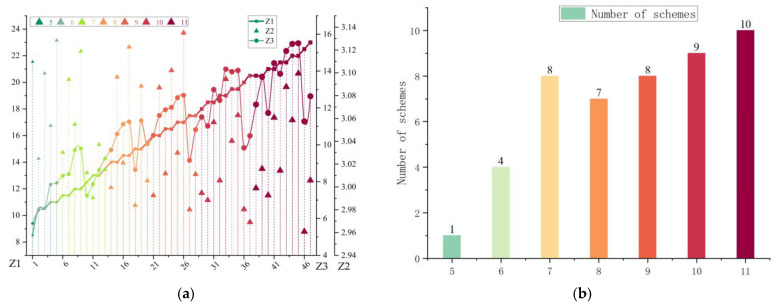
Analytical diagram of the objective function values for the site selection options. Where (**a**) shows the trend of the change of the objective value for each site selection scheme, and (**b**) shows the number of Pareto solutions for the different number of site selection points.

**Figure 12 biomimetics-08-00254-f012:**
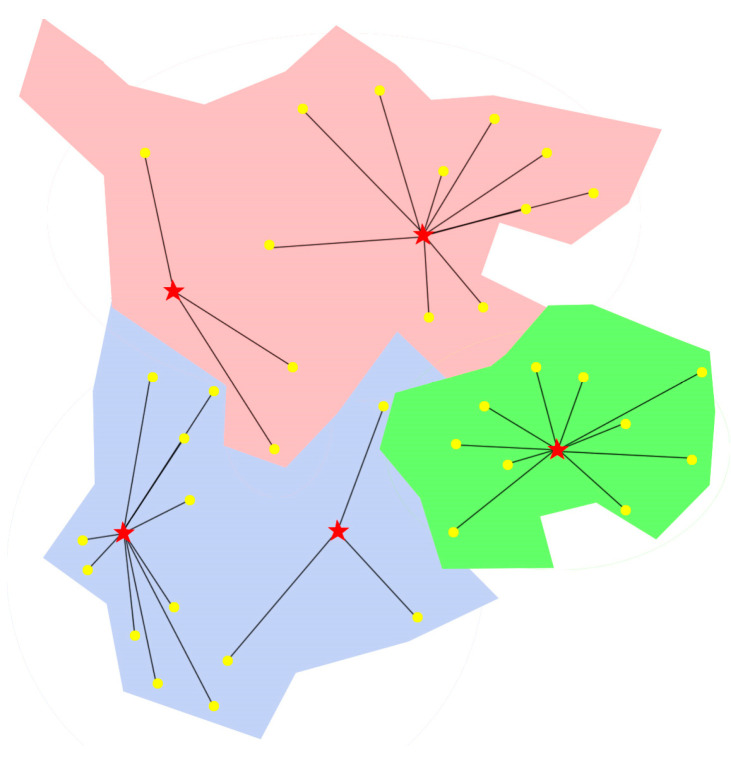
The radiation effect chart of the results of the five center sites. Where the different colors indicate different administrative regions, and the asterisk point indicates the final regional aviation emergency rescue center site selection point.

**Figure 13 biomimetics-08-00254-f013:**
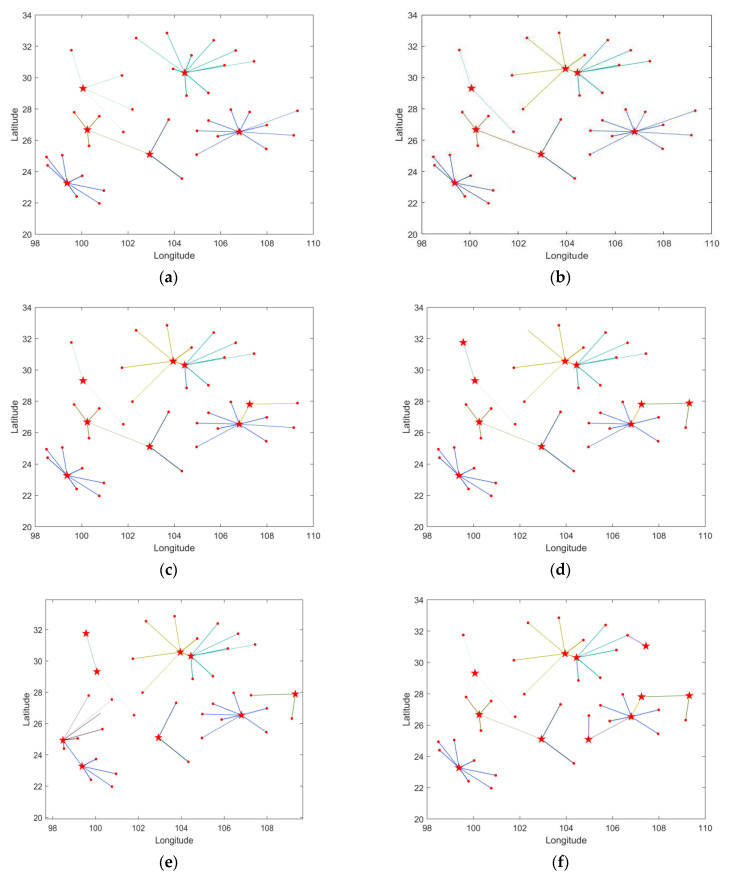
The number of different site selection points priority construction cost site selection results map site selection scheme latitude and longitude coordinates map. (**a**–**f**) show the latitudinal and longitudinal coordinates of the site selection scheme for 6, 7, 8, 9, 10, and 11 sites, respectively, obtained by processing with MATLAB software.

**Table 1 biomimetics-08-00254-t001:** Table of radiation coverage in a certain area.

$\varepsilon_{i}$	Node	$\tau_{ij}$	$\varepsilon_{i}\cdot \tau_{ij}$	$\varepsilon_{i}$	Node	$\tau_{ij}$	$\varepsilon_{i}\cdot \tau_{ij}$
0.836	V_4_	0.786	6.57096	0.545	V_7_	0.343	1.86935
0.836	V_5_	0.322	2.69192	0.545	V_8_	0.712	3.8804
0.836	V_6_	0.336	2.80896	0.326	V_12_	0.283	0.92258
0.836	V_8_	0.124	1.03664	0.326	V_11_	0.452	1.47352
0.836	V_9_	0.833	6.96388	0.326	V_13_	0.251	0.81826
0.836	V_11_	0.116	6.78832	0.326	V_14_	0.097	0.31622
0.836	V_10_	0.812	0.96976	Total	$Z_{3}$		37.11

**Table 2 biomimetics-08-00254-t002:** Table of coefficients of the model objective function.

Node	*A_i_*	$\varepsilon_{i}$	Node	*A_i_*	$\varepsilon_{i}$	Node	*A_i_*	$\varepsilon_{i}$	Node	*A_i_*	$\varepsilon_{i}\;$
V_1_	1	0.994	V_12_	2	0.78	V_23_	2.5	0.762	V_34_	1	0.926
V_2_	2.5	0.694	V_13_	2	0.702	V_24_	2	0.811	V_35_	1	0.938
V_3_	2.5	0.716	V_14_	2.5	0.707	V_25_	2.5	0.691	V_36_	2	0.815
V_4_	2	0.762	V_15_	2.5	0.7	V_26_	2.5	0.512	V_37_	2.5	0.689
V_5_	2.5	0.693	V_16_	2	0.786	V_27_	2.5	0.603	V_38_	2.5	0.677
V_6_	2.5	0.705	V_17_	2.5	0.682	V_28_	2.5	0.536	V_39_	2.5	0.615
V_7_	2.5	0.68	V_18_	2.5	0.725	V_29_	2	0.732	V_40_	2.5	0.623
V_8_	2.5	0.731	V_19_	2.5	0.703	V_30_	2.5	0.653	V_41_	2.5	0.582
V_9_	2.5	0.713	V_20_	2.5	0.701	V_31_	2.5	0.702	V_42_	2.5	0.568
V_10_	2.5	0.728	V_21_	1.5	0.895	V_32_	2.5	0.693			
V_11_	2.5	0.714	V_22_	2.5	0.722	V_33_	1	0.926			

**Table 3 biomimetics-08-00254-t003:** Radiance table for A region.

$\tau_{ij}$	V_1_	V_2_	V_3_	V_4_	V_5_	V_6_	V_7_	V_8_	V_9_	V_10_	V_11_	V_12_	V_13_	V_14_	V_15_
V_1_	1.000	0.715	0.394	0.000	0.000	0.000	0.309	0.000	0.000	0.000	0.535	0.360	0.000	0.279	0.630
V_2_	0.715	1.000	0.123	0.000	0.000	0.000	0.000	0.000	0.000	0.000	0.000	0.000	0.000	0.000	0.000
V_3_	0.394	0.123	1.000	1.000	0.940	0.830	0.916	0.445	0.272	0.457	0.440	0.000	0.000	0.000	0.000
V_4_	0.000	0.000	1.000	1.000	0.981	0.764	0.762	0.258	0.000	0.104	0.000	0.000	0.000	0.000	0.000
V_5_	0.000	0.000	0.940	0.981	1.000	1.000	0.912	0.637	0.474	0.521	0.271	0.000	0.000	0.000	0.000
V_6_	0.000	0.000	0.830	0.764	1.000	1.000	1.000	0.896	0.770	0.796	0.572	0.000	0.000	0.000	0.000
V_7_	0.309	0.000	0.916	0.762	0.912	1.000	1.000	0.837	0.751	0.854	0.757	0.433	0.000	0.000	0.000
V_8_	0.000	0.000	0.445	0.258	0.637	0.896	0.837	1.000	1.000	1.000	0.687	0.471	0.000	0.000	0.000
V_9_	0.000	0.000	0.272	0.000	0.474	0.770	0.751	1.000	1.000	1.000	0.747	0.599	0.368	0.117	0.000
V_10_	0.000	0.000	0.457	0.104	0.521	0.796	0.854	1.000	1.000	1.000	0.924	0.752	0.493	0.428	0.000
V_11_	0.535	0.000	0.440	0.000	0.271	0.572	0.757	0.687	0.747	0.924	1.000	0.993	0.671	0.754	0.000
V_12_	0.360	0.000	0.000	0.000	0.000	0.000	0.433	0.471	0.599	0.752	0.993	1.000	0.935	0.998	0.000
V_13_	0.000	0.000	0.000	0.000	0.000	0.000	0.000	0.000	0.368	0.493	0.671	0.935	1.000	0.972	0.000
V_14_	0.279	0.000	0.000	0.000	0.000	0.000	0.000	0.000	0.117	0.428	0.754	0.998	0.972	1.000	0.408
V_15_	0.630	0.000	0.000	0.000	0.000	0.000	0.000	0.000	0.000	0.000	0.000	0.000	0.000	0.408	1.000

**Table 4 biomimetics-08-00254-t004:** Partial table of Pareto solution results for different site points.

Number of Sites	Z1	Z2	Z3	Site Selection Results	Number of Sites	Z1	Z2	Z3	Site Selection Results
5	8.5	3.1096	5.73	1,10,21,29,34	9	16	3.0871	11.589	1,6,9,16,21,25,29,34,35
6	11	3.1284	7.939	1,6,9,21,29,34	9	17.5	3.0111	10.819	1,6,12,16,18,21,29,35,38
6	11	3.0537	7.846	1,3,9,21,29,34	9	16	2.9926	10.511	1,6,9,16,18,21,29,34,35
6	10.5	3.0995	6.547	1,6,12,21,29,34	9	17.5	2.9802	9.159	1,6,12,16,18,21,25,29,34,35
6	10.5	3.0248	6.454	1,3,12,21,29,34	10	19	3.0945	14.104	1,6,9,16,18,21,23,29,34,35
7	12	3.1189	9.803	1,6,9,21,29,34,35	10	19.5	3.0629	14.036	1,3,6,9,18,21,23,29,34,35
7	12	3.0549	9.71	1,3,9,21,29,34,35	10	19.5	3.0404	13.968	1,6,12,16,18,21,27,29,35,38
7	13.5	3.0152	9.253	1,6,9,18,21,29,34	10	18.5	3.0568	12.996	1,6,9,16,18,21,25,29,34,35
7	13	3.0371	8.632	1,6,12,18,21,29,34	10	19	3.0057	12.415	1,6,8,9,17,18,21,29,34,35
7	11.5	3.0942	8.411	1,6,12,21,29,34,35	10	18	2.9946	11.511	1,6,12,16,18,21,27,29,34,35
7	11.5	3.0302	8.318	1,3,12,21,29,34,35	10	18.5	2.9884	11.023	1,6,9,16,18,21,27,29,34,35
7	13	2.9905	7.861	1,6,9,16,21,29,34	10	20.5	2.9691	10.488	1,6,12,16,18,21,27,29,32,34,35
7	12.5	3.0124	7.24	1,6,12,16,21,29,34	10	20	2.9804	9.836	1,6,12,18,21,22,27,29,34,38
8	15	3.0884	11.315	1,3,9,18,25,29,34,35	11	22	3.0995	15.511	1,3,6,8,9,10,18,21,29,34,35
8	14.5	3.1227	11.247	1,6,9,18,21,29,34,35	11	22	3.0586	15.477	1,6,12,16,18,21,22,27,29,35,38
8	14.5	3.0211	11.117	1,6,12,16,18,21,29,34	11	21.5	3.0877	15.089	1,3,6,9,16,18,21,23,29,34,35
8	14	3.0963	10.589	1,6,12,18,21,29,34,35	11	21	3.0608	14.434	1,6,12,16,18,21,23,27,29,34,35
8	15.5	3.0055	10.039	1,6,12,16,18,21,29,34,35	11	21.5	3.0143	13.853	1,3,6,8,9,18,21,23,29,34,35
8	14	2.9994	9.725	1,3,9,16,21,29,34,35	11	20.5	3.0159	13.683	1,6,12,16,18,21,25,29,34,35,38
8	15	2.9839	8.647	1,6,9,16,18,21,29,34	11	23	3.0058	12.635	1,6,9,16,17,18,21,27,29,35,38
9	17	3.1350	12.688	1,3,6,9,18,21,29,34,35	11	20.5	2.9989	12.188	1,6,9,12,16,18,21,25,29,34,35
9	17	3.0299	12.555	1,6,12,16,18,21,27,29,34	11	21	2.9926	11.725	1,6,9,16,18,21,23,27,29,34,35
9	16.5	3.1021	12.033	1,6,9,16,21,23,29,34,35	11	22.5	2.9609	11.274	1,6,9,16,18,21,22,27,29,35,38
9	16.5	3.0118	11.903	1,6,9,18,21,23,29,34,35					

**Table 5 biomimetics-08-00254-t005:** The radiation table of the results of the five center sites.

Regional Center Point	Construction Cost	Radiation Point	Radiation Intensity	Regional Center Point	Construction Cost	Radiation Point	Radiation Intensity
1	1	2	0.71094	21	1.5	19	0.835035
		3	0.391298			20	0.567248
		15	0.626637			22	0.777097
10	2.5	4	0.075634			23	0.703301
		5	0.379504			24	0.895
		6	0.579144			25	0.570339
		7	0.621756			26	0.758657
		8	0.728	34	1	28	0.481791
		9	0.728			32	0.81762
		11	0.67252			33	0.772973
		12	0.547789			35	0.926
		13	0.358833			36	0.888472
		14	0.311779			37	0.736854
29	2.5	27	0.317826			38	0.345245
		30	0.313934			39	0.509028
		31	0.145489			40	0.567362
21	1.5	16	0.421572			41	0.499575
		17	0.787821			42	0.358454
		18	0.815967				

## Data Availability

The datasets used in the present study are available from this first author upon reasonable request.

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
