# Peer review of "A Study on Site Selection for Regional Air Rescue Centers Based on Multi-Objective Jellyfish Search Algorithm"

_biomimetics, 2023, doi:10.3390/biomimetics8020254_

Round 1

Reviewer 1 Report

First of all I will say that It is very gratifying to review proposals that do not fall into the field of connectionism and rescue the study and application of bioinspired metaheuristics.

Although the issue of rescue centers may be a bit foreign to many western regions, the issue is clearly described and the proposal is methodologically strong.

My only comment concerns the following subject:

  The proposed multi-objective function considers a set of conditions to perform optimization, but curiously does not take into account simpler issues such as the quality, experience or recent activity of the rescuers considered in the available crew rotation.

No major issues detected.

Reviewer 2 Report

The paper proposes a new approach based on Multi-Objective Jellyfish Search Algorithm used for location selection of regional aviation emergency rescue centers by considering three factors: construction cost, radiation range, and response time. The proposed method and the obtained results are interested, but the paper must be carefully revised for the sake of clarity and accuracy.

The following issues are recommended to improve the paper:

1.     Abstract: besides others (paper objective, methods, results, main conclusions), please also address here briefly the context for your research and paper novelty.

2.     Define all acronyms & symbols at their first use in the body text, even they are well known in literature. E.g., GIS, AHP, FA, GSA, ABC, DE, PSO, GA, etc.

3.     Avoid repeating information: the section of lines 58-73 is repeated (identically and in other words) into the line 74-93.

4.     Introduction: typically, this section ends by stating explicitly the paper novelty addressed in the paper, i.e., emphasizing the specific new / innovative aspects, considering actual knowledge at worldwide level. Also, the paper structure should be detailed according to its Sections.

5.     A generalized flowchart of the proposed approach is welcomed.

6.     civil aviation airports and A1 general airports” – please explain “A1” or cite a relevant document.

7.     Define tij and V in Eq. (6). Generally, define all symbols used in equations. Use Eq. (7) instead of “Formula 7”. If an equation is known in literature (e.g., Eq, (8)?), please cite de appropriate work. Use Eq. (7) and  Eq. (11) instead of “Equation 7 and Equation 11”.

8.     Use Figure 3 (or Fig. 3) instead of “the above figure”. Use Figure 8 instead of “”the figure below”. Indicate the parts: A, B, and C into Figure 8 (or state Vi…Vj for each region).

9.     Figures 4 and 5: branches of Yes and No should be discriminate!

10.  Line 474: explain in the paper the choice of η = 4 and why J0 Ï{0 0.25 0.75 1} .

11.  When the number of iterations does not satisfy the maximum number of iterations, the optimization search needs to continue.” Superfluous general statement!

12.  A Nomenclature of the used acronyms & symbols is welcomed.

13.  Typically, the last Section is named “Conclusion”, while Discussion are associated to the Results.

14.  Few typing mistakes should be solved, e.g. use space between values and its measurement unit.

Round 2

Reviewer 2 Report

Optional recommendation to include a statement on the main objective of each section of the paper, starting with the Section 2, at the end of the Section 1. Introduction.